# VLA-4 Expression and Activation in B Cell Malignancies: Functional and Clinical Aspects

**DOI:** 10.3390/ijms21062206

**Published:** 2020-03-23

**Authors:** Andrea Härzschel, Antonella Zucchetto, Valter Gattei, Tanja Nicole Hartmann

**Affiliations:** 1Department of Internal Medicine I, Medical Center and Faculty of Medicine, University of Freiburg, 79106 Freiburg, Germany; andrea.haerzschel@uniklinik-freiburg.de; 2Clinical and Experimental Onco-Hematology Unit, Centro di Riferimento Oncologico di Aviano (CRO) IRCCS, 33081 Aviano, Italy; zucchetto.soecs@cro.it (A.Z.); vgattei@cro.it (V.G.)

**Keywords:** lymphoma, leukemia, tumor microenvironment, integrin, B cell differentiation, adhesion, B cell receptor, therapy, Bruton’s tyrosine kinase, CD49d, chronic lymphocytic leukemia, CLL

## Abstract

Lineage commitment and differentiation of hematopoietic cells takes place in well-defined microenvironmental surroundings. Communication with other cell types is a vital prerequisite for the normal functions of the immune system, while disturbances in this communication support the development and progression of neoplastic disease. Integrins such as the integrin very late antigen-4 (VLA-4; CD49d/CD29) control the localization of healthy as well as malignant B cells within the tissue, and thus determine the patterns of organ infiltration. Malignant B cells retain some key characteristics of their normal counterparts, with B cell receptor (BCR) signaling and integrin-mediated adhesion being essential mediators of tumor cell homing, survival and proliferation. It is thus not surprising that targeting the BCR pathway using small molecule inhibitors has proved highly effective in the treatment of B cell malignancies. Attenuation of BCR-dependent lymphoma–microenvironment interactions was, in this regard, described as a main mechanism critically contributing to the efficacy of these agents. Here, we review the contribution of VLA-4 to normal B cell differentiation on the one hand, and to the pathophysiology of B cell malignancies on the other hand. We describe its impact as a prognostic marker, its interplay with BCR signaling and its predictive role for novel BCR-targeting therapies, in chronic lymphocytic leukemia and beyond.

## 1. Integrins in the Hematopoietic System

The communication between hematopoietic cells and their microenvironment in primary and secondary lymphoid organs is relevant for the functioning of immune cells, and disturbances in this communication are characteristic of hematologic neoplasia. B cell malignancies can arise from any stage of B cell differentiation and the malignant clones usually still contain characteristics of the cell-of-origin. Therefore, understanding homeostasis is a prerequisite for understanding and successfully treating cancer.

In health, B cell development and differentiation occur in well-defined sequential steps. The initial, antigen-independent stage, which comprises the differentiation from pro-B cells via pre-B cells and immature B cells to transitional (mature) B cells, takes place in the bone marrow. B cells then leave the bone marrow at the transitional B cell stage and complete the antigen-independent maturation into immunocompetent naïve mature B cells in the spleen. Upon antigen-binding and co-stimulation, further B cell differentiation takes place in secondary lymphoid organs.

During these differentiation steps, B cells rely on adhesive mechanisms. First, extravasation, tissue entry and retention are vital processes during the development and selection of B cells. Second, the interactions of B cells with other cell types, such as antigen-presenting cells (APCs) and T cells, require cell–cell contact. One of the most important families of cell adhesion receptors that mediate cell–cell and cell–extracellular matrix interactions is the integrin family. The term integrin stems from the capacity of these molecules to bi-directionally propagate signals across the cell membrane, thereby integrating signals from the extracellular environment into cytoplasmic signaling. Integrins are heterodimeric molecules of two non-covalently associated transmembrane subunits, the alpha and beta chains, and are classified on the basis of the combination of the alpha and beta subunit. In mammals, 24 possible heterodimers have been identified, deriving from differential combination of 18 α subunits and eight β subunits (reviewed, e.g., in [1], Scheme 1A). The α4 subunit can couple with either β7 or β1 subunits. The integrin very late antigen-4, VLA-4 (α4/β1, in other terms CD49d/CD29) is primarily expressed on leukocytes and best studied in the context of its role as a key mediator of hematopoietic stem- and progenitor cell homing and retention in bone marrow. The other α4 containing integrin, α4/β7 orchestrates T cell migration to the intestine by binding to its ligand MAdCAM-1 [2], and will therefore not be addressed in the following chapters.

While VLA-4 is the dominant integrin in hematopoietic progenitors, B cells express two major integrins, namely VLA-4 and lymphocyte function-associated antigen 1 (LFA-1, αLβ2). The usage and function of these integrins depend on the differentiation stage of the B cells. VLA-4 has emerged early during evolution and can contribute to the functions of B cells that are related to innate immune responses, e.g., T-independent antibody responses. LFA-1, which arose only in the last part of vertebrate evolution, is crucial to adaptive functions, e.g., the positioning of B cells in secondary lymphoid organs for T–B cell interactions [3,4]. Nevertheless, in the adaptive context, VLA-4 is involved in the acquisition of antigen by B cells and their subsequent activation [5,6]. VLA-4 also contributes to leukocyte extravasation to secondary tissue sites during inflammation, which is a multistep process. Thereby, VLA-4 has not only the capacity to mediate the typical integrin-dependent late steps of strong adhesion to the endothelium, which are determined by high-affinity interactions with the ligand, but also to orchestrate low-affinity rolling on the endothelium, which is an earlier step of the cascade and classically attributed to selectins. These characteristics are related to structural features, most importantly the composition of domains in the α4 subunit. The molecular structure of VLA-4 is shown in Scheme 1B. VLA-4 affinity can be dynamically upregulated by a process called inside-out activation, which is best characterized in the context of the chemokines presented by the endothelial cells (for details, see Section 5).

Major VLA-4 ligands are the extracellular matrix protein fibronectin and VCAM-1. VCAM-1 expression is constitutively present on various stromal cells and endothelial cells, but upregulated by inflammatory signals in a NF-kB-dependent manner [7]. Besides these ligands, certain forms of osteopontin and other non-classical ligands such as emilin have been suggested as VLA-4 partners [8,9,10]. Moreover, VLA-4 was suggested to interact with JAM-B, a junctional molecule, during the step of actual transmigration [11].

Although the importance of VLA-4 for stem- and progenitor cells in the bone marrow, and thus for the maintenance of a healthy immune system, has been acknowledged for decades, new aspects of its functioning in different hematopoietic cell types are still being discovered. This is mainly due to its complex levels of regulation, which allow for a multitude of cell type- and context-dependent functions. In this review, we first describe the major importance of VLA-4 in the development and maturation of the healthy B cell pool. We then focus on B cell malignancies, describing what is known about the prognostic value of VLA-4 as well as the mechanisms of its action under pathologic conditions. Finally, we examine its therapeutic impact as a predictive factor under BCR inhibitor therapy and possible future roles as a therapeutic target.

## 2. VLA-4 Functions during B Cell Development in Bone Marrow

B cell development in the bone marrow is VLA-4-dependent and involves their adhesive capacity at several stages. Early studies using chimeric mice have reported that mice with a deficiency in the alpha subunit of VLA-4 (CD49d) generate very few B cells due to impaired B cell differentiation at the pro-B cell stage [12,13]. It is also an early finding that B cell precursors, during their development, rely on the VLA-4-dependent binding to VCAM-1 presented by stromal cells [14,15]. This might not only confer their appropriate localization towards supportive signals such as CXCL12, but also lead to the initiation of signal transduction from the pre-BCR. An in vitro study reported the formation of an immune synapse between human pre-B and stromal cells. Into this synapse, the pre-BCR is recruited through the cross-linking of pre-B cell integrins by stromal ligands to initiate pre-BCR signaling [16]. There is also interesting evidence that the adhesion of pre-B lymphoblastic cells to stroma is a biphasic process, with VLA-4 being the dominant player of the very first phase of adhesion, whereas later phases of retention are controlled by other players [17].

Although fibronectin, which can bind both VLA-4 and VLA-5, is highly abundant in bone marrow, the VLA-4-specific ligand VCAM-1, rather than fibronectin, orchestrates the adhesive interactions of B cell precursors to the bone marrow microenvironment [14]. Data from in vivo homing experiments also support a primary role of the VCAM-1-VLA-4 rather than the fibronectin-VLA-5 axis in the pathophysiology of precursor-B acute lymphoblastic leukemia (ALL) cells [18]. However, in vitro findings indicate that fibronectin also has the capacity to enhance the proliferation of certain malignant pro-B cells [19]. In any case, high VLA-4 expression is associated with adverse outcome and distinct gene expression changes in childhood B-cell precursor ALL at first relapse [20]. Whether or not there is a therapeutic window for blocking VLA-4 in this disease remains to be elucidated. In vitro, a protective role of stromal cells towards leukemia cells can be observed under chemotherapy, e.g., cytarabine, and VLA-4 inhibition was observed to abolish this protection [20]. However, not only childhood ALL is characterized by VLA-4 expression. For example, unusually high VLA-4 expression was observed in a case of an adult with aleukemic B-cell ALL presenting with osteolytic bone lesions, and was particularly prominent in the osteolytic regions of the patient, in conjunction with a high expression of VCAM-1, CXCL12 and CXCR4 [21]. This observation suggests a role of these molecules in malignant bone remodeling.

During their differentiation towards the immature and transitional stage, developing B cells require a dynamic retention to the bone marrow parenchyma, which includes adhesive and de-adhesive processes. Indeed, the immature B cell stage is characterized by a significant change in positioning within bone marrow, with up to 50% of immature B cells localizing within sinusoids in a VLA-4-dependent manner [22]. To adapt to their localization, the B cells migrate in an amoeboid way, regulated by the interplay of CXCR4 and VLA-4, expressed on their surface with the respective ligands CXCL12 and VCAM-1, expressed by the environment. The chemokine axis serves to activate the VLA-4 integrin and to guide the direction of the actual migration. The role of the activated VLA-4 is to withstand the mechanical shear forces in the microenvironment. Its dynamic activation and de-activation allow the appropriate balance of retention and migration. For example, VLA-4-mediated adhesion is temporally reduced when cells are in perisinusoidal compartments, before it is increased again through cannabinoid receptor 2-induced transactivation within sinusoids to prevent premature cell egress from bone marrow [23].

## 3. Marginal Zone B Cells and Other Mature B Cell Subsets

Once out of the bone marrow, B cell differentiation continues in distinct zones of the spleen. Splenic marginal zone B cells exhibit unique functional characteristics because they contribute to innate immune responses, in addition to their participation in T-cell-dependent immune responses by importing blood-borne antigens to follicular areas of the spleen. This means they can mount a local antibody response against type-2 T-cell-independent (TI-2) antigens and they shuttle between the blood-filled marginal zone for antigen collection and the follicle for antigen delivery. This shuttle is regulated by differential usage of the integrins LFA-1 and VLA-4, with VLA-4 being responsible for adhesion and migration down the flow, while interacting with VCAM-1 [24]. Another highly integrin-dependent B cell subset populating similar splenic niches to marginal zone B cells are memory B cells, which, in consequence, also rely on high integrin expression for their proper localization and function [25].

The importance of integrins in mediating the precise localization of B cells within the different splenic compartments is also evident under pathologic conditions. In a murine chronic lymphocytic leukemia (CLL) model, which is driven by the proliferation of malignant B cells in follicles, the inhibition of VLA-4-mediated adhesion and CXCL13-mediated follicular homing displaced leukemic cells not only from the follicle, but also the marginal zone, and reduced leukemia progression [26,27]. Consistently, in B and T cell lymphoma 3D models, VLA-4 was critical for maintaining the adhesion of the lymphoma cells to follicular dendritic cells. This interaction upregulated the expression levels of the B cell receptor, which again supported the survival of lymphomas through a tyrosine kinase Syk in the upstream BCR pathway [28]. It is likely that marginal zone-specific factors shape the phenotype of leukemic cells and facilitate their niche-specific retention. In this context, the homeobox protein NKX2-3, which acts as an oncogene, promotes marginal zone-lymphomagenesis by activating B-cell receptor signaling [29]. This, in turn, activates relevant adhesion molecules such as VLA-4 and CXCR4, in a Lyn/Syk-dependent way, eventually driving malignant transformation through triggering NF-κB and PI3K-AKT pathways. Another regulator of marginal zone B cell development from T1 to T2 transitional B cells is SWAP-70, a Rho GTPase-interacting and F-actin-binding protein with functions in cell polarization, migration, and adhesion, thus regulating marginal zone B development and marginal zone formation [30]. SWAP-70 acts as a negative regulator of integrin-dependent adhesion and is particularly important for the differentiation control of B-cell precursors and their contribution to splenic tissue formation [30]. A positive regulator of activation of VLA-4 and other integrins, talin, is not required for follicular B-cell maturation in the spleen or homeostatic humoral immunity, but was found to be critical for integrin-dependent B lymphocyte emigration to lymph nodes and for optimal immunity against T-dependent antigens [31].

The central nervous system (CNS) represents an exceptional challenge for the immune system due to the blood–brain barrier, which restricts and regulates the access of immune cells to this organ. B cells can cross the blood–brain barrier in an integrin-dependent manner to ensure CNS immune surveillance [32,33], and thus it is not surprising that they were also shown to contribute to autoimmune disease involving the CNS, most prominently to multiple sclerosis (MS). Specifically, B cell recruitment to the CNS was dependent on VLA-4 and neutrophil-derived CXCL27 [34]. In line with this, the specific deletion of CD49d on B cells decreased their capacity to infiltrate the CNS and improved outcome in experimental autoimmune encephalitis (EAE), a mouse model of MS. Decreased B cell infiltration went along with a decrease in other leukocyte subsets, specifically macrophages and Th17 T cells [35], suggesting a major role for B cells in the recruitment of these cell types. On the other hand, regulatory B cells were also affected by CD49d deletion in the EAE model [36], a potentially counterproductive effect in the treatment of inflammatory diseases which may, however, be beneficial in other settings involving immune suppression, such as certain cancers.

## 4. The Peculiar Connection of B Cell Activation and VLA-4 in Chronic Lymphocytic Leukemia

VLA-4 is involved in the acquisition of antigen by B cells and their subsequent activation, lowering the activation threshold [4,6]. This is achieved by strengthening the adhesive connections between antigen-presenting cells and B cells, which facilitates antigen uptake. However, the amplification of BCR signaling via integrin outside-in signals may also play a role. In particular, it has been shown that the co-expression of VCAM-1 on antigen-bearing membranes enhances antigen-dependent B-cell activation [5]. Firstly, this is achieved by increasing the adhesion strength of the B cell to the antigen-presenting substrate by the additional VLA-4-VCAM-1 ligation, which, in turn, increases the likelihood of a B cell to be activated. Secondly, the VLA-4-VCAM-1 binding supports the docking structure characteristic of the B-cell immunological synapse and thereby strengthens the bi-cellular interactions of the antigen-presenting and antigen-recognizing cell partner [5], resulting in enhanced BCR signaling. This promoting effect of VLA-4 on the BCR is particularly effective when the affinity of the BCR for the antigen is low. The other way around, BCR activation also alters integrin-mediated adhesiveness, which is useful during differentiation and maturation because the requirements for specific adhesive interactions in this process change during the sequential steps. Spaargaren and colleagues proposed that BCR ligation induces VLA-4 activation by a signaling cascade involving PI3K, BTK, PLCγ2 and calcium mobilization. Finally, this cascade leads to calpain-mediated release of VLA-4 molecules from cytoskeletal constraint and consequent cluster formation and increased adhesion [37].

In contrast to normal mature B cells, which constitutively express VLA-4, its expression is absent or low on approximately 50% of CLL cases. CD49d expression in CLL is epigenetically driven and associated with certain types of genetic lesions, like trisomy 12 [38] and Notch1 mutations [39]. Overall, CD49d expression on more than 30% of CLL cells represents a robust negative prognostic marker for treatment-free and overall survival [40,41,42,43].

Recently, we reported that BCR-engagement induces VLA-4 activation in CLL. CLL is a malignancy of mature B cells that harbor a characteristic CD5 expression as well as memory features. The malignant B lymphocytes proliferate in lymphoid organs and require signals from this peculiar microenvironment to maintain the disease (for review, see [44]). The considerable therapeutic potential in disrupting these CLL cell–microenvironment interactions and the need to understand the molecular basis of these signaling axes is underscored by recent clinical developments. One important example is ibrutinib, an orally administered covalent inhibitor of BTK, which had been approved in 2014 for CLL patients who have received at least one previous therapy, and is meanwhile approved for all CLL indications due to its remarkable clinical efficacy [45]. From the point of view of biological signaling cascades, BTK can be located downstream of BCR engagement but also downstream of other microenvironment-relevant receptors, such as toll-like receptors or adhesion molecules. A characteristic capacity of ibrutinib is to dislocate leukemic cells from the lymphoid organs into the periphery, so that the patients present with a transient leukocytosis. We observed that VLA-4 activation interferes with this characteristic [46]. This means that CD49d expression has also a predictive role for inferior patient outcome under ibrutinib [43,46]. CD49d-positive and CD49d-negative CLL patients, i.e., harboring VLA-4 or not on their leukemia cells, were followed under ibrutinib treatment. In CD49d-negative patients treated with ibrutinib, a rapid reduction in lymphadenopathy was observed, paralleled by a transient lymphocytosis. However, this pattern was less prominent in CD49d-positive CLL samples. Mechanistically, this observation was due to the residual inside-out activation of VLA-4 by BCR stimulation in lymphoid organs [46]. The data imply that BTK inhibition might not be sufficient to block BCR-dependent VLA-4 activation and to induce a strong initial lymphocytosis in this patient cohort, which further influences therapy outcome. In other words, the signaling axis from the BCR towards the VLA-4 integrin might be able to bypass BTK. Of interest, BCR-induced inside-out activation of the second major lymphocyte integrin, LFA-1, has been reported to occur through a BTK-independent pathway involving phosphatidylinositide 3-kinase (PI3K) [47]). Because we observed residual phosphorylation of the downstream PI3K target AKT in ibrutinib-treated CLL cells upon BCR stimulation, we subsequently studied the effects of combined BTK and PI3K inhibition on BCR-induced VLA-4 activation and could confirm an involvement of PI3K in this pathway [46].

In a follow-up study, this was further preclinically validated. By using the TCL1-tg mouse model, which resembles the clinical CD49d high-risk group for CLL, PI3K, rather than BTK, was found as an essential part of the signaling between the BCR and VLA-4 [48], supporting the observation made in humans. Furthermore, therapeutically targeting VLA-4 in this mouse model in a transplantation setup confirmed the important role of this molecule for CLL pathogenesis and resulted in a reduced tumor load in lymphoid organs of the treated mice. The data suggest that in future clinical trials and real-world clinical application, the monitoring of CD49d expression should be incorporated to further elucidate and confirm these findings. In long-term, VLA-4 inhibition or the additional targeting of BCR pathway molecules (e.g., by other inhibitors of PI3K than idelalisib to avoid its severe side effects [49]) might be explored as additions to ibrutinib therapy for high-risk CD49d-high CLL patients.

## 5. The Inside-Out VLA-4 Activation Cascade in Detail

The data outlined above suggest VLA-4 as a potential therapeutic target structure for the treatment of B cell malignancies. In this context, it has to be considered that integrin function depends on a complex activation process that involves other cellular receptors and signaling pathways.

VLA-4 activation has two dimensions, “inside-out” and “outside-in”. The term inside-out signaling encompasses a number of possible pathways and processes, with several basic mechanisms in common. It starts with the activation of a cell surface receptor, e.g., a cytokine-, chemokine- or antigen receptor, initiating an intracellular signaling cascade, the core elements of which are PI3K and PLCγ (Scheme 2). These signaling events entail conformational changes in the integrin molecule as well as the altered spatial distribution of integrin molecules on the cell, which regulates avidity. The various outcomes concerning integrin conformation and -distribution can occur separately or interdependently, depending on the cellular and microenvironmental context, yet the result is always increased cellular adhesiveness. Among the relevant activation pathways in B cells, chemokine-induced VLA-4 activation is the most rapid and involves conformational changes towards higher affinity within fractions of a second. BCR-induced integrin activation is somewhat slower and longer lasting. Within seconds/minutes, integrin clustering can be observed, and this enhances the avidity of ligand binding. Differences in integrin activation also lie in the different kinetics of activation. The α4 subunit of VLA-4 does not contain a regulatory I-domain, which results in a much faster kinetics of ligand binding compared to α-I-domain-containing integrins such as LFA-1. In consequence, VLA-4 (in contrast to LFA-1) can mediate selectin-independent rolling on VCAM substrates, and firm arrest to VCAM-1 under shear flow is possible even in the absence of inside-out activation, albeit at low levels [50]. On a biochemical level, the overall binding potential of a cell to VLA-4 ligands is determined by the speed at which receptor–ligand bonds form and dissociate (kinetics) and by the lifetime of those bonds (thermodynamic equilibrium).

To alter the equilibria of bond formation and dissociation, the VLA-4 integrin can adopt several different conformations, depending on a) molecular extension b) the affinity of the ligand binding pocket. Inside-out activation goes along with profound changes in the arrangement of all domains of the integrin molecule. Ligand binding then induces further conformational changes, with the consequence of hybrid domain exposure, which has thus been proposed as an easily detectable surrogate marker for ligand binding [51] (Scheme 3). However, there are also hints that hybrid domain movement may precede and facilitate the rearrangement of the βA domain, and thus the high-affinity conformation of the ligand-binding pocket [52]. The exact molecular mechanisms of activation can differ between integrins and depends on the cellular context.

Methods to detect the variable conformation states include the use of FRET probes at the integrin head and the plasma membrane to detect bended or extended conformations [53]. In the specific case of VLA-4, the affinity of the binding pocket can be determined by measuring the binding kinetics of a synthetic small molecule ligand based on the LDV sequence of fibronectin [54], and hybrid domain exposure can be detected using the monoclonal antibody HUTS-21 [51,55]. Combining these tools revealed that VLA-4 molecules occur on resting lymphocytes mainly in a low-affinity bent confirmation with a hidden hybrid domain. Inside-out signals via G-protein–coupled receptors (GPCRs) induce both extension and affinity upregulation, while ligand-binding induces hybrid domain exposure. However, extension and affinity upregulation can also take place independently of each other. PMA, an activator of the protein kinase C (PKC)-Rap axis, induces high-affinity integrins without causing extension. On the other hand, the high ligand affinity induced by GPCR signaling is rapidly desensitized, while the VLA-4 extended form prevails for much longer [53]. On a functional level, extended integrins enable the fast formation of receptor–ligand bonds, while the stability of those bonds depends on the affinity state [3] (Scheme 4A,B). Of note, this applies to physiological, surface-bound ligands only, while the binding of monovalent, small molecule ligands such as LDV is exclusively dependent on regulation of the dissociation rate by ligand affinity [54,56].

Besides the conformation of single integrin molecules, the adhesive properties of a cell are further modulated by their spatial distribution on the cell surface. The overall avidity of an integrin-expressing cell to an integrin-ligand presenting surface can thus be enhanced by the formation of adhesive spots via integrin clustering [57] (Scheme 4C). This requires cytoskeletal re-organization enabling the lateral mobility of the molecules on the cell surface, and can happen in addition to or independent of the conformational changes. Experimentally, integrin clustering can be detected using confocal microscopy. The physiological activation of integrins via chemokine receptors commonly engages all of the above mentioned mechanisms, as reviewed by Laudanna et al. [58]. Which of the mechanisms described are initiated by a certain activating stimulus, however, is highly dependent on the specific integrin, the cell type on which it is expressed and other conditions such as the activation or metabolic state of the cell. Furthermore, the exact signaling pathways leading to the different outcomes of inside-out activation are highly context-specific.

In experiments using cells from Waldenström macroglobulinemia (WM, a lymphoplasmacytic lymphoma) patients, ibrutinib and idelalisib both display inhibitory characteristics towards BCR-controlled integrin-mediated adhesion, whereas chemokine (CXCL12/CXCR4)-controlled signaling, adhesion and migration are not affected by drugs blocking BTK and PI3K [59]. In the CLL context, it was hypothesized that only chemokines, but not BCR signals, trigger JAK protein tyrosine kinases that boost full BTK activity in terms of integrin activation [60]. This suggests a selective use of the JAK-BTK axis upstream of integrins, which is dependent on the microenvironment signal. Focusing on the second lymphocyte integrin LFA-1, rather than on VLA-4, and not stratifying the used patient samples for CD49d (VLA-4) expression in this study, the authors also reported that BCR-induced LFA activation did not involve affinity regulation, but rather integrin clustering (avidity regulation). In the case of VLA-4, we have observed both avidity and affinity regulation upon BCR engagement in CLL [46]. These processes might differently be stabilized by separate upstream kinases. For example, by using a VLA-4-positive CLL-derived cell line model and genetic modulation of ZAP-70, Laufer et al. recently reported that integrin avidity regulation upon chemokine stimulation involved ZAP-70 expression, whereas high-affinity regulation of integrins was independent of ZAP70 [61]. An exact understanding of the involved kinases upstream of VLA-4 and how these kinases are triggered by the various stimuli remains to be elucidated.

Avidity regulation might also take place downstream of initial VLA-4 inside-out activation. The stabilization of integrin activation requires the binding of intracellular adaptor molecules, such as talins and kindlins, to the short intracellular part of the β subunit. This, in turn, enables the linkage of the integrin to the actin cytoskeleton, thus enabling force-induced adhesion strengthening. In experiments using CLL and multiple myeloma cell lines, both tumor entities shared a dependence on kindlin-3 and talin, and both adaptors cooperatively stimulated a high-affinity and strength of VLA-4-dependent attachment to bone marrow endothelium [62]. Integrin Cytoplasmic domain-Associated Protein-1 (ICAP-1), a specific adaptor of the β1 integrin subunit cytoplasmic domain, was described as a negative regulator of adhesion in this study. In another study, the elevation of cytoplasmic cyclic nucleotides was suggested as another main mechanism of decreasing VLA-4 activation [63]. It will be interesting to elucidate the association of these individual components.

Overall, an assessment of the activation state of VLA-4, rather than its mere surface expression, may help to gain prognostic and functional information. For therapeutic intervention, upstream pathways like PKC or PI3K are considerable targets. Direct targeting of the integrin may overrule complexity from upstream inside-out activation, but has some drawbacks, as will be outlined in the next paragraph.

## 6. Lessons Learned from Natalizumab

A therapeutic potential of VLA-4 inhibition in cancer could be most reasonable as a combination approach. VLA-4 inhibition will most likely dislocate CD49d-positive tumor cells from bone marrow and lymphoid organs, and thereby improve the effectiveness of the combination drug or even overcome drug resistance. Most therapeutic approaches focus on targeting the CD49d (α4) subunit, leading to the inhibition of both VLA-4 and α4β7. Therapeutic options range from antisense approaches and small molecule antagonists to antibody therapies. A limitation of VLA-4 as a therapeutic target is its broad expression on immune cells. Indeed, use of the monoclonal humanized IgG4 anti-CD49d antibody Natalizumab (tysabri) requires a careful risk management. Natalizumab has been developed in the context of inflammation-mediated diseases and autoimmune conditions and is currently used for the treatment of MS [64] and Crohn’s disease [65]. In patients treated with Natalizumab [33], impaired immune surveillance of the CNS can be observed, along with an increased risk of opportunistic infections. Indeed, among the adverse effects occurring mainly after long-term treatment, is progressive multifocal leukoencephalopathy (PML), a rare but severe and often fatal condition caused by John Cunningham (JC)-virus infection [66]. This resulted in a voluntary withdrawal of the drug from the market in 2005. Remarkably, advocacy groups for patients lobbied the FDA to make this drug available again due to its unique efficacy (reviewed in [67]) and Natalizumab was re-approved in 2006. Nowadays, a detailed standardized risk management program is established, including comprehensive laboratory testing and immune monitoring of complete blood count, peripheral immune cell status, and serological parameters, as well as JC-virus testing, before the start of therapy. Leukocytosis is monitored due to the mobilizing potential of VLA-4 inhibition. Indeed, hematopoietic stem cells are mobilized from the bone marrow by Natalizumab, with high levels of circulating stem- and progenitor cells maintained over time [68,69]. Of note, Natalizumab-intrinsic side effects might be related not only to the broad immune cell expression of VLA-4, but also to pharmacokinetics, i.e., with a high stability and plasma half-life of 16 ± 4 days after infusion [70]. This characteristic could be ameliorated using small molecule inhibitors with faster pharmacokinetics (half-life of about 3–5 h). Among the developed inhibitors, firategrast (e.g., NCT00395317) is the most advanced, with the completion of phase I and II trials. Firategrast was observed to reduce disease activity, but to a lesser extent than Natalizumab. The further development of this inhibitor might benefit from different formulations. AJM300 (INN; Carotegrast methyl) is another small-molecule α4-integrin antagonist that can be administered orally and is almost non-immunogenic. The previous results of a phase II study on ulcerative colitis have been encouraging [71] and, currently, a Phase III Study is recruiting (NCT03531892). In addition, an antisense therapy to CD49d (ATL1102) has recently brought encouraging results in a phase II trial in patients with Duchenne Muscular Dystrophy (clinical trial registry number ACTRN12618000970246, active trial).

Whether or not VLA-4 targeting substances will find their way into the treatment of hematological diseases is not yet clear. Successful clinical development will require a reduced risk and toxicity profile of the inhibitors that allows the combination of VLA-4 antagonists with other drug with a high efficacy of the drug. If this is achieved, not only patients with autoimmune- and other inflammatory diseases, but also those with hematologic malignancies, could greatly benefit from VLA-4 inhibition.

## 7. Conclusion and Future Perspectives of VLA-4 as a Therapeutic Target

VLA-4 has been known for a long time as an essential homing and retention factor of hematopoietic stem- and progenitor cells. Its adhesive properties are modulated by a complex network of regulation, leading to a great functional versatility. This versatility presents a great challenge to researchers working on VLA-4, as its functions do not only depend on the cell type on which it is expressed and its developmental stage, but also on the composition of the environment regarding other cells, extracellular matrix elements and soluble factors. This needs to be considered when drawing conclusions from data acquired from isolated cells, which, in the case of leukemia patients, are usually derived from peripheral blood. This insight, together with valuable data gained from more systemic analyses that are possible in animal models, may reveal more functions of VLA-4 and further enhance its prognostic and predictive potential for B cell- and other malignancies in the future. A better understanding on VLA-4 biology and/or more specific inhibitors blocking defined functions may also be a means to enhance the value of VLA-4 as a direct therapeutic target.

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
