# Peer review of "VLA-4 Expression and Activation in B Cell Malignancies: Functional and Clinical Aspects"

_ijms, 2020, doi:10.3390/ijms21062206_

Round 1

Reviewer 1 Report

In their review article entitled “VLA-4 expression and activation in B cell malignancies: functional and clinical aspects”, Andrea Härzschel and colleagues describe the role of VLA-4 in B-cell differentiation and in haematological malignancies. This topic is not only academically interesting but has also profound implications with respect to homing of malignant cells and the advent of novel therapies that change mobility of malignant cells. The review therefore is of great interest to the community.

Major points

  • More graphs would be nice:
    • Overview figure of integrin families / structure would be helpful to the integrin-naïve reader
    • g. to illustrate the connection of different signaling pathways (inside-out, outside-in, BCR-VLA4-PI3K)
  • The description of binding affinity, extension and hybrid domain on page 6 bottom third could be improved. It remains unclear (at least to me as naïve reader) what the function of the hybrid domain is and the function of the bending. How does bending influence affinity? Is there a connection between bent high affinity form and extended high affinity?
  • The section "The inside-out VLA-4 activation cascade in detail“somewhat lacks context with the preceding and following section. The concept of “Inside-out regulation” could be explained in more detail for the naïve reader.
  • What is the hybrid domain and what is its function? (maybe a short explanation 1-2 sentences)
  • Scheme 2:
    • in the figure legend to A it says that affinity is modulated. However, in panel A this is not intuitively shown?
    • Panel B should there be a difference in affinity = stars between extended and bent? This is not intuitive to me. The panel is about extension, but extended and bent can bind so the actual message could be more focused? Text and image seem slightly off?
    • Same in Panel C: should there be a difference between left and right in affinity?
  • The description of malignancies is somewhat biased towards CLL.

Minor points

  • Title 1. Introduction and 7. Conclusion could be less generic and more descriptive of the content of these passages.
  • In the introduction sometimes sentences are not well directly connected to each other, making it difficult to understand their relevance and relation during a „first-read“
  • Line 124: Confused about which is the ligand of VLA-4, it says the ligand is VLA-4 in the microenvironment?
  • Resolution of figures is too low
  • Editing by a native speaker could be beneficial,
    • g. changing word order line 310: “…are not affected by drugs blocking BTK and PI3K”: descriptions should precede nouns.
    • “the” frequently missing, e.g. line 38 “in the bone marrow”, line 40 “in the spleen”, line 302 “mechanisms as reviewed by Laudanna” without comma, line 314 “rather than on”, line 343 “approaches and small molecular antagonists to”, line 375 “is not yet clear”, 384 “expressed as well as”, 387 “This insight together”  s.f.

Author Response

In their review article entitled “VLA-4 expression and activation in B cell malignancies: functional and clinical aspects”, Andrea Härzschel and colleagues describe the role of VLA-4 in B-cell differentiation and in haematological malignancies. This topic is not only academically interesting but has also profound implications with respect to homing of malignant cells and the advent of novel therapies that change mobility of malignant cells. The review therefore is of great interest to the community.

We thank the reviewer for the positive assessment and valuable input. Based on the reviewer’s comments we have adapted the manuscript as follows:

Major points

  1. More graphs would be nice:
    • Overview figure of integrin families / structure would be helpful to the integrin-naïve reader
    • g. to illustrate the connection of different signaling pathways (inside-out, outside-in, BCR-VLA4-PI3K)

As suggested by the reviewer, we have added two new schemes. New Scheme 1 gives an overview of the integrin family and the structure of VLA-4, and new Scheme 2 depicts the most important inside-out and outside-in signaling pathways and their major molecular players.

  1. The description of binding affinity, extension and hybrid domain on page 6 bottom third could be improved. It remains unclear (at least to me as naïve reader) what the function of the hybrid domain is and the function of the bending. How does bending influence affinity? Is there a connection between bent high affinity form and extended high affinity?

We thank the reviewer for this valuable comment, and have extensively revised the first part of Chapter 5 to strengthen its clarity, particularly for readers not specialized in integrins. Specifically, we have added a section on the hybrid domain (lines 295-301 of the revised manuscript). Further, we have expanded the figure legend of new Scheme 4 (formerly Scheme 2), to clarify that extension and affinity upregulation are separate processes, although they often occur in parallel.

  1. The section "The inside-out VLA-4 activation cascade in detail “somewhat lacks context with the preceding and following section. The concept of “Inside-out regulation” could be explained in more detail for the naïve reader

To strengthen the contextual clarity and flow of the manuscript, we have added new paragraphs in the beginning (lines 269-272 of the revised manuscript) and at the end (lines 374-377 of the revised manuscript) of the respective chapter. We have also added the new Scheme 2 to illustrate inside-out signaling, with concomitant edition of respective passages (lines 273-277 of the revised manuscript).

  1. What is the hybrid domain and what is its function? (maybe a short explanation 1-2 sentences)

See response to 2.

  1. Scheme 2:
    • in the figure legend to A it says that affinity is modulated. However, in panel A this is not intuitively shown?
    • Panel B should there be a difference in affinity = stars between extended and bent? This is not intuitive to me. The panel is about extension, but extended and bent can bind so the actual message could be more focused? Text and image seem slightly off?
    • Same in Panel C: should there be a difference between left and right in affinity?

We edited the figure legend and the corresponding text section. We hope we could clarify that the three processes of affinity upregulation, extension and clustering are independent processes, although they can occur simultaneously (new figure legend of Scheme 4).

  1. The description of malignancies is somewhat biased towards CLL.

Indeed, the CLL “bias” is due to the number of papers published. A relatively large number of studies on functional and clinical aspects of VLA-4 (CD49d) expression have been conducted in CLL. As of current knowledge, CLL is the B cell malignancy, in which the predictive and prognostic role of CD49d expression is best established. In other entities, less is known yet but the importance of VLA-4 might be explored in future. To point out the CLL focus, we have modified the abstract (line 26-27).

Minor points

  • Title 1. Introduction and 7. Conclusion could be less generic and more descriptive of the content of these passages.

The sections were renamed (lines 32 and 419 of the revised manuscript)

  • In the introduction sometimes sentences are not well directly connected to each other, making it difficult to understand their relevance and relation during a „first-read“

The whole section has been edited for coherence.

Line 124: Confused about which is the ligand of VLA-4, it says the ligand is VLA-4 in the microenvironment?

We apologize for the mistake, which was corrected (new line 148).

  • Resolution of figures is too low

We have improved the resolution of the figures.

  • Editing by a native speaker could be beneficial,
    • g. changing word order line 310: “…are not affected by drugs blocking BTK and PI3K”: descriptions should precede nouns.
    • “the” frequently missing, e.g. line 38 “in the bone marrow”, line 40 “in the spleen”, line 302 “mechanisms as reviewed by Laudanna” without comma, line 314 “rather than on”, line 343 “approaches and small molecular antagonists to”, line 375 “is not yet clear”, 384 “expressed as well as”, 387 “This insight together”  s.f.

We have corrected all these issues.

Reviewer 2 Report

This is a well written review and acceptable for publication in its current form. Only one minor correction: line 124, third word "VLA-4" should be "VCAM-1".

Author Response

  • This is a well written review and acceptable for publication in its current form. Only one minor correction: line 124, third word "VLA-4" should be "VCAM-1".

We thank the reviewer for the positive assessment and apologize for the mistake, it was corrected.